# A Sustainable Strategy for Solid-Phase Extraction of Antiviral Drug from Environmental Waters by Immobilized Hydrogen Bond Acceptor

**DOI:** 10.3390/nano12081287

**Published:** 2022-04-10

**Authors:** Hongrui Yang, Chen Wang, Wenjuan Zhu, Xia Zhang, Tiemei Li, Jing Fan

**Affiliations:** School of Environment, Key Laboratory of Yellow River and Huai River Water Environment and Pollution Control, Ministry of Education, Henan Key Laboratory for Environmental Pollution Control, Henan Normal University, Xinxiang 453007, China; yangyangyang321200@163.com (H.Y.); wang2119084002@163.com (C.W.); wenjuanz87@126.com (W.Z.); zhangxia27@126.com (X.Z.); lucy-limei@163.com (T.L.)

**Keywords:** immobilized hydrogen bond acceptor, separation and enrichment, arbidol, environmental waters

## Abstract

Deep eutectic solvents are a new generation of green solvents composed of hydrogen bond acceptors and donors. However, when used as extractants in liquid–liquid separation, they are difficult to recycle and easy to lose. In order to solve these problems, herein, immobilized hydrogen bond acceptor adsorbent material was prepared for the separation and enrichment of antiviral drug arbidol from seven kinds of environmental water samples by in situ formation of hydrophobic deep eutectic solvents. The structure, morphology and thermal stability of the adsorbents were characterized, the separation and enrichment conditions for the targeted analyte were optimized, and the adsorption thermodynamics and kinetics were investigated. It was found that the adsorbent material could effectively enrich trace arbidol with the recovery more than 95% at the concentration above 7.5 ng/mL, and the enrichment factor was as high as 634.7. Coexisting substances, such as NaCl, KCl, CaCl_2_ and MgCl_2_, did not interfere with the adsorption of arbidol, even if their concentration was high, up to 1.0 mol/L, and the relative recovery for real samples was in the range from 92.5% to 100.3%. Furthermore, the immobilized hydrogen bond acceptor could be recycled and reused, and the recovery of arbidol was still above 95% after 12 adsorption–desorption cycles. The mechanism study demonstrates that the synergistic effect of hydrogen bonding and π-π stacking is the primary factor for the high adsorption efficiency.

## 1. Introduction

Since the year 2019, the novel coronavirus (COVID-19) has quickly spread across the world and posed a serious threat to public health because it can infect people very easily [1]. Owing to its extremely effective antiviral capability, arbidol has recently been recommended to treat COVID-19 for improving the discharging rate and decreasing the mortality rate [2]. However, the maximum conversion rate of arbidol in the human body has been found to be 60%, with the untransformed residues excreted in feces [3]. The widespread use of arbidol would lead to a large amount of arbidol in the natural environment, which poses a potential risk to the environment and human health [4]. A literature survey reveals that the studies on the determination and removal of arbidol in water environments have not been reported up to date, thus it is immensely necessary and meaningful to develop a material that can effectively separate and enrich arbidol from contaminated water.

Deep eutectic solvents (DESs) are a class of green and eco-friendly solvents, comprised of hydrogen bond acceptors (HBAs) and hydrogen bond donors (HBDs) at a certain stoichiometric ratio [5,6]. Owing to their attractive characteristics, such as simple preparation procedure, low price, high atomic utilization rate and good biocompatibility [7], DESs have been successfully applied in the fields of extraction of organic chemicals [8,9,10,11], microextraction of analytes [12,13], sample preparation [14,15], biomass dissolution [16], catalytic reactions [17,18] and electrochemistry [19,20]. However, most DESs reported previously were hydrophilic, thus greatly limiting their applications in aqueous medium. Even though the hydrophobic deep eutectic solvents have been quickly developed since 2015 to meet this challenge [21], the use of hydrophobic DES_S_ in aqueous phase extraction also has problems, such as easy loss of DES [22], difficult recovery of the analyte, and hard recycling of the adsorbent [23,24].

Inspired by the concept of immobilized ionic liquids [25,26,27], immobilized hydrogen bond acceptor (immobilized-HBA) was proposed in this work to address the problems in the application of hydrophobic DESs. For this purpose, 2-dimethylaminoethanol was grafted on the chloromethyl polystyrene resin by chemical bonding to form immobilized-HBA, which was then applied for solid-phase extraction of the antiviral drug arbidol (HBD) from contaminated environmental waters by the formation of deep eutectic solvent with the drug to achieve the purpose of enrichment and separation of the pollutant. It was found that this strategy had a simple separation procedure, and exhibited high adsorption efficiency. Combined with simple ultraviolet spectrophotometry, arbidol can be directly detected at the level of ng/mL. In addition, the immobilized-HBA could be easily regenerated and reutilized, and almost no wastes were discharged in the whole of sample processing. The established method was used for the separation and analysis of arbidol in environmental waters with the advantages of simple operation procedures, high sensitivity, good selectivity, no need for expensive instruments, and environmentally friendly processes.

## 2. Experimental

### 2.1. Reagents and Materials

Chloromethyl polystyrene resin (PS-CH_2_Cl, 1% crosslinked polystyrene, 200–400 mesh, chlorinity 3.5 mmol/g, Tianjin Nankai Hecheng Sci. & Tech. Co., Ltd., Tianjin, China), 2-dimethylaminoethanol (99%, AR, Aladdin), arbidol (>98%, AR, Aladdin), *N*-methyl-2-pyrrolidone (99%, AR, Zhengzhou Paini Chemical Reagent Factory, Zhengzhou, China), acetic anhydride (AR, Tianjin, Damao Chemical Reagent Factory, Tianjin, China), choline chloride (99%, AR, Shanghai Macklin Biochemical Co. Ltd., Shanghai, China), and absolute ethanol (AR, Shanghai, Wokai Biotechnology Co. Ltd., Shanghai, China) were used in our experiments, except choline chloride, which was dried under vacuum at 60 °C for 12 h before use, all the other chemicals were used as received.

### 2.2. Apparatus

Infrared (IR) spectrometer (FTS-40, Bole Co., Ltd., Los Angeles, CA, USA), field emission scanning electron microscopy (FE-SEM, Hitachi Hi-Tech Co., Ltd., SU8010, Tokyo, Japan), integrated thermal gravimetric analyzer (TGA, Netzsch Co., Ltd., STA449C, Selb, Munich, Germany), UV–vis spectrophotometer (Beijing, PERSEE General Instrument Co., Ltd., TU-1810, Beijing, China), differential scanning calorimetry (DSC, 204F1, Selb, Germany), solid phase extractor (Tianjin hengao technology development Co., Ltd., HSE-12B, Tianjin, China), solid phase extraction (SPE) column with 10.0 mm in diameter and 62 mm in length, and magnetic stirrer (Henan aibote technology development Co., Ltd., CJB-S-5D, Zhengzhou, China).

### 2.3. Preparation of the Immobilized Hydrogen Bond Acceptor

The immobilized HBA was synthesized by following the procedure displayed in Figure 1, which is the modification of a previously reported procedure [28]. Briefly, 0.5 g of chloromethyl polystyrene resin was swelled in 20 mL of *N*-methyl-2-pyrrolidone for 12 h, 10 mL of 2-dimethylaminoethanol was then added, and the mixture was heated at 45 °C for 12 h under magnetic stirring with the speed of 360 rpm/min. The target product was washed by distilled water and ethanol and then dried under vacuum at 45 °C for 12 h.

### 2.4. Solid-Phase Extraction Experiments and the Determination of Arbidol

The adsorption of arbidol by the immobilized-HBA was studied by batch experiments and dynamic column adsorption tests. In order to evaluate the adsorption performance of the immobilized-HBA material, the pH value was adjusted by acetic acid solution (0.1 mol/L), and the UV-visible absorbance of arbidol solution was detected at λmax = 316 nm for both batch experiments and column experiments. All the determinations of the solution samples were conducted in triplicate.

#### 2.4.1. Batch Adsorption Tests

Batch adsorption tests were adopted to evaluate the adsorption performance of arbidol by the immobilized-HBA. In doing so, adsorption conditions were optimized and adsorption isotherms were determined. Unless it is specifically requested, all the adsorption tests were carried out in a series of identical sample bottles with 10 mg of the immobilized-HBA and different concentrations of arbidol. Afterwards, the mixture of the immobilized-HBA and arbidol was magnetically stirred for 30 min at 20 °C. The immobilized-HBA was sunk to the bottom of the system, thereafter, the concentration of arbidol in the supernatant was measured. The adsorption capacity and recovery rate of arbidol were calculated by the following equations:(1)$$q_{e}=\frac{(C_{0}-C_{e})V}{m}$$
(2)$$q_{t}=\frac{(C_{0}-C_{t})V}{m}$$
(3)$$R\%=\frac{\left(C_{0}-C_{e}\right)}{C_{0}}\times 100\%$$
where *q_e_* and *q_t_* are the equilibrium adsorption capacity and the adsorption capacity at contact time t; *C*_0_ (mg/mL), *C_e_* (mg/mL) and *C_t_* (mg/mL) are the concentrations of arbidol at the beginning, equilibrium and time t, respectively; *m* (mg) is the mass of the adsorbent, *V* (mL) is the volume of arbidol solution, and R% stands for the recovery efficiency.

#### 2.4.2. Column Extraction Tests

Column adsorption tests were conducted to study the influence of sample volume, interfering substances and adsorbent regeneration. Herein, 100 mg of the immobilized-HBA was encapsulated in a solid phase extraction (SPE) column, the arbidol solution was flowed through the column with the optimized speed (3.75 mL/min). Arbidol reserved on the column was eluted by ethanol, and then measured by spectrophotometry. The relative recovery (%) was figured up as follows:(4)$$Recovery\%=\frac{q_{eluted}}{q_{added}}\times 100\%$$
where *q_eluted_* (μg) stands for the mass of eluted arbidol from the column of immobilized-HBA, and *q_added_* (μg) is the mass of arbidol added into the actual sample.

## 3. Results and Discussion

### 3.1. Characterization and Analysis of the Adsorption Materials

In order to investigate the morphology and element distribution of the PS-CH_2_Cl and the immobilized-HBA, the SEM and EDS mapping are of great importance. As shown in Figure 1a,b, there was no obvious difference in the morphology between PS-CH_2_Cl and immobilized-HBA, they were spherical with the size of 80–130 μm in the diameter. Besides, the EDS spectrum and the corresponding element content were shown in Appendix A, which were applied to indicate the elemental mapping images of PS-CH_2_Cl and immobilized-HBA. The results indicated that the new characteristic peaks of oxygen and nitrogen appeared in the immobilized-HBA after 2-dimethylaminoethanol was grafted on PS-CH_2_Cl. Compared with PS-CH_2_Cl, the content of carbon and chlorine of the immobilized-HBA decreased, demonstrating that 2-dimethylaminoethanol was successfully decorated onto the surface of the PS-CH_2_Cl.

In order to examine the formation of immobilized-HBA, IR spectrum is an attractive method to obtain relevant information. FT-IR spectra of PS-CH_2_Cl and immobilized-HBA were shown in Appendix A. The peaks at 1450 cm^−1^, 2920 cm^−1^ and 3025 cm^−1^ were attributed to the skeleton vibration peaks of C-C (benzene ring), C-H (methylene) and C-H (benzene ring) of chloromethyl polystyrene resin and immobilized-HBA [29]. The peaks at 1086 cm^−1^ and 3235 cm^−1^ were ascribed to the stretching vibration of C-N and O-H of 2-dimethylaminoethanol. Moreover, the characteristic peaks at 671 cm^−1^ and 1264 cm^−1^ were, respectively, the stretching vibration and bending vibration of C-Cl in the PS-CH_2_Cl rather than in the immobilized-HBA. The new peak at 1477 cm^−1^ (C-N stretching vibration) was a special peak of quaternary ammonium salt [30], which is strongly confirmed that 2-dimethylaminoethanol was successfully grafted onto the PS-CH_2_Cl and is also consistent with the result reported in literature [28].

Thermogravimetric analysis (TGA) was used to study the thermal stability of PS-CH_2_Cl and immobilized-HBA under the atmosphere of N_2_ and to estimate the content of 2-dimethylaminoethanol grafted onto the chloromethyl polystyrene resin. It was clearly seen from Appendix A that chloromethyl polystyrene resin was stable below 300 °C, and then its stability gradually declined in 300–550 °C with a weight loss of 78.6%. While in the temperature range of 100–200 °C, the immobilized-HBA had a slight weight loss, which might be attributed to the water loss. Surprisingly, the immobilized-HBA exhibited a remarkable weight loss of 92.4% in the temperature range of 200–480 °C. Taking all these data into account, it is clear that the immobilized-HBA was stable up to 200 °C and the content of 2-dimethylaminoethanol immobilized on the surface of PS-CH_2_Cl was about 14.2%. Furthermore, all the results of SEM, EDS, IR and TGA demonstrated that the 2-dimethylaminoethanol was grafted on the PS-CH_2_Cl without changing its morphology.

In addition, considering the fact that the structure of choline chloride is similar to that of the immobilized-HBA, we chosen choline chloride as HBA and arbidol as HBD to study the formation of DES. The general procedures for the preparation of the deep eutectic mixture of choline chloride and arbidol was shown in Appendix A. Importantly, as shown in Appendix A, a new peak at 127 cm^−1^ was observed after mixing arbidol and choline chloride uniformly. This new peak could be attributed to the hydrogen bonding of O-H…Cl [31]. At the same time, the peak at 1067 cm^−1^ for the stretching vibration of arbidol phenolic C-OH [32] was shifted to 1086 cm^−1^ (Appendix A), also indicating that there was a strong hydrogen bonding between choline chloride (Cl^−^) and arbidol (-OH) [31]. Additionally, after arbidol and choline chloride were combined by hydrogen bonds, the melting point (88.4 °C) of the DES was much lower than that of choline chloride (305.1 °C) and arbidol (134.8 °C), as shown in Appendix A. All these results indicate that choline chloride and arbidol formed DES by hydrogen bonding, which is the indirect but useful evidence for the formation of DES from the immobilized-HBA and arbidol.

### 3.2. Effects of pH and Temperature

Solution pH is of utmost significance in the process of adsorption. In view of this, solution pH in the range of 3–9 was chosen to examine the adsorption of arbidol on the immobilized-HBA. For the sake of comparison, adsorption performance of the carrier PS-CH_2_Cl for arbidol was also investigated (in Figure 2). It was found that the blank carrier PS-CH_2_Cl did not adsorb arbidol, which may be attributed to its non-porous structure, but the immobilized-HBA did have high adsorption efficiency, indicating that the active site of the adsorbent was on the immobilized-HBA but not on the blank carrier PS-CH_2_Cl. It is known that pKa = 6.5 for arbidol in aqueous solution. According to the ion/molecule fraction distribution formula (Equation (S1)) [33], the ionization degree of arbidol at different pH values was calculated, and the result was shown in Appendix A. It can be seen that when pH < 6, arbidol mainly exists in molecular form, which is good for the formation of hydrogen bonds between arbidol and the immobilized-HBA. However, at pH < 5, a large number of H^+^ cations in acidic aqueous solution can electrostatically interact with Cl^−^ of the immobilized-HBA, which is not favorable to the hydrogen bonding between the hydrogen bond donor and acceptor. On the other hand, it can be seen from Appendix A that when pH = pKa (6.5), about 50% arbidol exists in the ionic form, while 75.9~99.7% arbidol exists in the ionic form in the pH range from 7 to 9. In other words, H proton of arbidol was gradually removed with increasing pH. In this case, arbidol gradually lost its ability to form hydrogen bonds with the immobilized-HBA, resulting in the reduced adsorption efficiency of arbidol. Therefore, the extraction efficiency is higher at pH 5–6, and the subsequent experiments were performed at the pH value of around 5.5.

Temperature is also of great significance for the affinity of the adsorbent towards arbidol. Therefore, the impact of temperature on the adsorption capacity was examined from 10 to 60 °C, which was shown in Appendix A. Due to the affinity between the adsorbent and arbidol, the adsorption process is exothermic, which means that increasing temperature is not conducive for the adsorption of arbidol. On the other hand, hydration of arbidol is much stronger in aqueous solutions at lower temperatures, which is also not beneficial for the adsorption of arbidol. Thus, higher adsorption efficiency was observed in the temperature range of 15–30 °C, and 20 °C was chosen for the next experiments.

### 3.3. Adsorption Isotherm

An adsorption isotherm study is useful for the study of the adsorption process and adsorption mechanism. As clearly shown in Appendix A, the adsorption capacity raised from 2.44 to 126.1 mg/g as the starting concentration of arbidol was increased from 5 to 300 mg/L. To investigate how the arbidol interacted with the adsorbent, the experimental results were analyzed by Langmuir and Freundlich isotherms according to Equations (5) and (6):(5)$$\frac{C_{e}}{q_{e}}=\frac{C_{e}}{q_{m}}+\frac{1}{q_{m}K_{L}}$$
(6)$$Inq_{e}=InK_{F}+\frac{1}{n}InC_{e}$$
where *q_e_* (mg/g) and *q_m_* (mg/g) are, respectively, the equilibrium adsorption capacity and the maximum adsorption capacity. *C_e_* (mg/L) stands for the equilibrium concentration of arbidol, and *K_L_* and *K_F_* are the constant of Langmuir and Freundlich models, respectively.

According to Equations (5) and (6), the possible linear Langmuir and Freundlich models were shown in Appendix A. It was observed that there was no good linear relationship between *C_e_/q_e_* and *C_e_*, which means that the adsorption isotherm could not be fitted by the Langmuir model. However, a good linear relationship between In *q_e_* and In *C_e_* was found, which demonstrates that heterogeneous adsorption sites and multilayer adsorption existed on the surface of the immobilized-HBA. These are consistent with the experimental results that the adsorption capacity increased with increasing initial concentration of arbidol.

### 3.4. Adsorption Kinetics

The adsorption kinetics are necessary to study the adsorption performance and mechanism. The adsorption kinetic curve of the adsorbent for arbidol was shown in Figure 3. It is clearly shown that the adsorption capacity (q_t_) increased quickly within 25 min, and then arrived at adsorption equilibrium, and the equilibrium adsorption capacity was 126.1 mg/g.

Next, the data were analyzed by the following pseudo-first-order model (Equation (7)) and pseudo-second-order model (Equation (8)) with the fitting curves as shown in Appendix A.
(7)$$In(q_{e}-q_{t})=Inq_{e}-k_{1}t$$
(8)$$\frac{t}{q_{t}}=\frac{1}{k_{2}q_{e}^{2}}+\frac{t}{q_{e}}$$
where *k*_1_ (min^−1^) and *k*_2_ (g/(mg·min)) are the rate constant of pseudo-first-order and pseudo-second-order adsorption, respectively, *q_t_* and *q_e_* (mg/g) have the same meaning as mentioned above. As shown in Equations (7) and (8), if these equations are applicable, *k*_1_, *k*_2_ and *q_e_* can be obtained through the slope and intercept of these liner equations [34]. The acquired fitting parameters were listed in Appendix A. It was noted from Appendix A, the adsorption process of arbidol perfectly fits the pseudo-second-order model with a quite high correlation coefficient (R^2^ = 0.994). Meanwhile, the calculated adsorption capacity of 145.8 mg/g was in good agreement with the experimental adsorption capacity of 126.1 mg/g.

### 3.5. Enrichment Efficiency of the Immobilized-HBA

Considering the low concentrations of contaminants in the environmental water samples, it is very important to evaluate the adsorption performance of adsorbent for low concentration pollutants. Compared with the batch process, the dynamic column adsorption has the advantage of high efficient utilization of the absorbability [35]. Thus, dynamic column adsorption was used to investigate the extraction performance of trace arbidol by packed immobilized-HBA. As listed in Table 1, the volume of arbidol solution at various concentrations were investigated in the range from 50 to 2000 mL by the dynamic column adsorption following the procedure described in Section 2.4.2. After optimizing the conditions, 3 mL of anhydrous ethanol was used to successfully elute the arbidol adsorbed on the column. The recovery of arbidol was still 95.2% even at low dosage of 0.0075 μg/mL (7.5 ng/mL) and the enrichment factor was high up to 634.7 as calculated from the ratio of the analyte concentration in eluent (C_f_) to that in initial sample solution (C_i_) [36]. Such a high enrichment factor is very beneficial to the remediation of slightly polluted water as well as the extraction and analysis of trace pollutant.

### 3.6. Anti-Interference Ability of the Immobilized-HBA

The complex matrix of natural waters may affect the adsorption of arbidol. In our work, the influence of common interfering substances in water was investigated. As shown in Table 2, when the concentration of common coexistence ions (such as K^+^, Na^+^, Ca^2+^, Mg^2+^, Cl^−^) in water was 1.0 mol/L, which is about 50,000 times higher than that of arbidol (2 $\times 10^{-5}\mathrm{mol}/L$), they still had barely impact on the extraction of arbidol within the error less than 2.7%. Additionally, compared with the effects of cations and monovalent anions, SO_4_^2−^ and PO_4_^3−^ had bigger impact on the adsorption of arbidol, however, the concentration of SO_4_^2−^ and PO_4_^3−^ in actual water samples was much lower than that tested in our experiments. Soluble starch and glucose have been studied as a representative of common coexisting organic compounds, even when the concentrations of glucose and soluble starch were 50,000 and 750 times that of arbidol, little effect was observed on the extraction efficiency. All these results demonstrate that the immobilized-HBA has a strong salt resistant, and is a strong candidate for the extraction of arbidol from high-salt wastewater.

### 3.7. Sustainability of Arbidol Wastewater Pretreatment by the Immobilized-HBA

Adsorbent regeneration is one of the most important indicators to evaluate the performance of adsorbents. At the same time, no waste discharge during sample processing is also the goal of sustainable and green sample processing. Therefore, it is of great significance for sustainable development to realize the recovery and reuse of eluent and arbidol in the process of immobilized-HBA regeneration. For this purpose, different eluents such as ethanol, acetic acid, the mixed solution of ethanol and acetic acid were selected to elute arbidol and regenerate the immobilized-HBA adsorbent. At the same time, the eluent dosage, eluent flow speed and elution recovery of arbidol were investigated, and the results were given in Table 3. It was found that ethanol was the best eluent, and the optimal elution efficiency was achieved by using 3 mL of eluent with the eluent flow speed of 0.2 mL/min. In addition, using ethanol as eluent has several advantages, for example, it can be performed under mild conditions, and it is easy to recycle, barely secondary pollution and has low toxicity.

The recycling and regeneration of arbidol and immobilized-HBA were researched. When the wastewater containing arbidol was flowed through the column filled with immobilized-HBA, arbidol was retained in the column. Then, arbidol could be eluted from the column by ethanol, and the immobilized-HBA was regenerated at the same time. After that, the eluted solution containing arbidol was distilled at about 78 °C, the ethanol was then collected and applied for next cycles, and arbidol in the eluent was recovered. It was found that after 12 adsorption-elution cycles, the recovery of arbidol by immobilized-HBA was still above 95%.

As shown in Appendix A, there was no significant difference in UV absorption between the original arbidol and the recovered arbidol at the same concentration, indicating that the purity of the recovered arbidol was high. The FT-IR spectra displayed in Appendix A confirm that the immobilized-HBA had excellent stability. In addition, after one-step treatment, the wastewater containing arbidol could be discharged to the standard, the adsorption material would be regenerated by ethanol eluent, the ethanol and arbidol could be separated by heating, and no waste was generated in the whole process. This environmental friendly and sustainable process is extremely value of application.

### 3.8. The Applications of Actual Water Samples

In order to verify its practicality, the immobilized-HBA was used to extract arbidol in many different kinds of actual water samples, such as tap water, rain water, river water, lake water, the Yellow River water, sewage effluents, and domestic sewage. At the same time, a given amount of arbidol was, respectively, added in the actual waters to prepare the samples with 0.05 μg/mL of arbidol for recovery experiments. All the actual water samples were filtered by a 0.22 μm of membrane before use, then arbidol content was determined by following the steps described in Section 2.4.2. In the column extraction tests, 300 mL of 0.05 μg/mL arbidol solution was passed through the column containing the immobilized-HBA with a flow speed of 3.75 mL/min, then the arbidol adsorbed on the immobilized-HBA was eluted with 3 mL of anhydrous ethanol at a flow speed of 0.2 mL/min. The arbidol concentration in the eluant was determined, and the results were shown in Table 4. It was found that the single-stage adsorption efficiency of arbidol in cleaner water was to exceed 96% owing to less interference. However, the adsorption efficiency of arbidol from domestic sewage and sewage effluents was declined with increased complexity degree, but it was still higher than 92%. Finally, a controlled experiment of arbidol before and after enrichment was performed (Figure 4). It was clearly demonstrated that arbidol in the unenriched sample could not be detected by a simple UV spectrophotometer. However, after being enriched, there was an obvious adsorption peak at λ = 316 nm, which could be accurately detected. All these results indicate that the immobilized HBA had a strong enrichment effect towards the trace target analyte, and is promising for arbidol extraction from intricate environmental water samples.

### 3.9. Adsorption Mechanism of the Immobilized-HBA

From the formation principle of DES, HBA and HBD are mainly bonded by hydrogen bonds. Combined with the conclusion mentioned in 3.2, arbidol (see molecular structure in Appendix A) mainly exists in the form of molecules when pH = 5.5. In this case, the H in hydroxyl group of arbidol may form hydrogen bond with Cl^−^ in the immobilized-HBA, leading to the in situ formation of hydrophobic DES. To prove the hydrogen bonding interactions, FT-IR spectra were determined and the results were displayed in Figure 5a. It can be seen from Figure 5(a1,a2) that there was no appreciable change for the peak of O-H stretching vibration (at 3235 cm^−1^) [37] of the immobilized-HBA before and after arbidol adsorption, indicating that the O-H played a negligible role in the formation of hydrogen bonds. In principle, chloride anions may have strong hydrogen bond interactions with a hydrogen bond donor [38,39]. In order to examine the formation of hydrogen bonds between the immobilized-HBA (Cl^−^) and arbidol (-OH), far infrared spectra were measured, which is sensitive to hydrogen bonding. As shown in Figure 5(b2), after arbidol was adsorbed by the immobilized-HBA, a new peak at 127 cm^−1^ was observed. This new peak might be attributed to the hydrogen bonding of O-H…Cl^−^ [31]. Moreover, the peak at 1067 cm^−1^ for the stretching vibration of arbidol phenolic C-OH [32] was shifted to 1088 cm^−1^ after adsorbed on the immobilized-HBA. These results indicate that hydrogen bonds were formed between chloride anions of the immobilized-HBA and the hydroxyl of arbidol.

According to the adsorption isotherms discussed above, the adsorption process of arbidol by immobilized-HBA conformed to Freundlich model and was a multi-layer adsorption, which could be ascribed to the π-π interaction between arbidol and immobilized-HBA because both the drug and the absorbent contain benzene rings. Moreover, as shown in the FT-IR spectra, the absorption peaks at 1687 cm^−1^ for the immobilized-HBA and 1687 cm^−1^ for the arbidol were attributed to phenolic skeleton vibration, and they shifted to 1616 cm^−1^ when the arbidol was adsorbed by the immobilized-HBA, manifesting that there might be π-π stacking between arbidol and the immobilized-HBA [35,37].

In addition, in order to further verify that there was hydrogen bonding but not ion exchange in arbidol adsorption, 1 mL supernatant was taken after arbidol adsorption by the immobilized-HBA, followed by adding one drop of aqueous HNO_3_ (0.1 mol/L) and then one drop of aqueous AgNO_3_ (0.1 mol/L). However, no AgCl precipitation was found, suggesting that no ion exchange appeared between the immobilized-HBA and arbidol. This was consistent with the fact that arbidol was present primarily in molecular form at pH = 5.5. Based on the above information, it is appropriate to state that DES was formed by hydrogen bonds between the hydroxyl group in arbidol and the chloride anion in the immobilized-HBA. Therefore, the synergistic effect of hydrogen bonding and π-π stacking is the main driving force for the efficient extraction of arbidol by immobilized-HBA. The possible adsorption mechanism is shown in Figure 6.

## 4. Conclusions

In summary, a highly effective adsorption material was synthesized though a one-step procedure by the chemical immobilization of hydrogen bond acceptor on the chloromethyl polystyrene resin for separation and analysis of arbidol from environmental waters. This strategy overcomes some disadvantages of the common DESs, such as the easy loss and difficult recycling of the adsorbent, as well as the complicated regeneration and recovery procedures of the target analytes. It is worth noting that within 25 min, the adsorption efficiency of arbidol from water reached more than 95% at the drug concentration above 7.5 ng/mL, with the enrichment factor up to 634.4. Such a striking adsorption performance was attributed to the formation of DES by hydrogen bonding between the immobilized-HBA and arbidol, together with the synergistic effect of π-π stacking. The adsorption isotherm could be fitted by Freundlich model, and the adsorption followed a pseudo-second-order kinetic equation. Furthermore, the adsorption material showed excellent salt resistance, even if the salt concentration was as high as 1.0 mol/L, which is about 50,000 times that of arbidol. This advantage was very beneficial to the advanced treatment of saline industrial wastewater, medical and domestic wastewater and other actual environmental waters. The immobilized-HBA was easy to regenerate and recycle, and no obvious decrease in the adsorption performance was observed after 12 cycles. At the same time, eluent and arbidol could be easily recovered and reused. Additionally, thus, the immobilized-HBA is a promising adsorption material to extract arbidol from the environmental samples. The established sample pretreatment method also provides a useful reference for sustainable green separation technology.

## Data Availability

Data is contained within the article and Appendix A.

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
