# Peer review of "A Sustainable Strategy for Solid-Phase Extraction of Antiviral Drug from Environmental Waters by Immobilized Hydrogen Bond Acceptor"

_nanomaterials, 2022, doi:10.3390/nano12081287_

Round 1

Reviewer 1 Report

Comments:

  1. Line 104 what does mean by HAc solution
  2. “Moreover, the characteristic peaks at 671 cm-1 and 1264 cm-1 were the vibration of C-Cl and CH2-Cl in the PS-CH2Cl rather than in the immobilized-HBA”. What does mean C-Cl ? It same as CH2-Cl.
  3. “All these results indicate that choline chloride and arbidol formed DES by hydrogen bonding, which is the indirect but useful evidence for the formation of DES from the immobilized-HBA and arbidol” Based on the procedure given in the supporting information for the synthesis of choline chloride arbidol DES, DES was formed through hydrogen bonding at 120 0C, but for the immobilized-HBA with arbidol, the adsorption done at 20 0 Can immobilized-HBA can able to form hydrogen bond with arbidol at 20 0C within 30 min? Here, immobilized-HBA has almost same structural base of choline chloride, please explain?
  4. “On the other hand, with the increase of pH value, the immobilized-HBA may react with excess OH- in alkaline environment to reduce the adsorption efficiency of arbidol” Please mention the pH from where to where. Give schematic explanation for structural changes of arbidol at different pH.
  5. Based on the Fig 2, extraction efficiency is higher at pH 5-6. In section 3.2, need more clear explanation about this.
  6. “The possible reason was that as the temperature went up, the water solubility of arbidol was increased, which could decrease the affinity between the immobilized-HBA and arbidol”. Based on this statement, what about the solubility of arbidol at 15-30 °C,
  7. In Table 1, What does mean Varbidol /mL in the first column
  8. In Table 2, authors mentioned about the soluble starch and glucose, while in section 3.6, these are not mentioning, please clarify.
  9. Please mention the experimental conditions for Table 1, Table 2, and Table 4
  10. Please check, Table 3 is missing
  11. Herein, different eluents such as ethanol, acetic acid, the mixed solution of ethanol and acetic acid were selected to elute and regenerate the immobilized-HBA adsorbent. The result showed that ethanol was the best eluent because of its high elution rate”. Please give more details about the results of this experiment such as ratio of ethanol and acetic acid, eluent rates, and regeneration %
  12. “All the actual water samples were filtered by a 0.22 mm of membrane, then pre-treated, and arbidol content was determined by following the steps described in section 2.4.2.” what does mean by pre-treated?
  13. All figures clarity needs to improve

Reviewer 2 Report

Authors investigated the use of immobilized hydrogen bond acceptor to recover arbidol from water samples. I recommend the article for publication after some changes:

Please replace the word synthesis when referring to DES. DES are not synthetized, there is not the formation of a new compound. DES are mixtures of compounds. This a common language mistake that must be eliminated.

What do authors mean with ‘low eutectic mixtures’ (page 2, line 45) and ‘the water instability of DESs’ (page 2, line 51)? These parts should be reformulated.

Table S2 is not mentioned in the manuscript.

Authors mention that chemicals were used as received. Choline chloride is extremely hygroscopic and its handling without special care leads to water absorption. The water content of this compound and its mixture with arbidol should be added; if significant, the effect of the presence of water in the themes approached should be explored

The verbal tenses and many words along the manuscripts are often wrong or inadequately used, what makes it hard to read. An extensive review must be performed.

Minor:

Line 91, page 2: two dots

Line 166, page 5: “analysis (TGA) was carried out” wrong verbal form

Line 213, page 6: “Carbidol = 25 mg/L,” arbidol should be placed below the line.

Line 342, page 9: abidol instead of arbidol

Round 2

Reviewer 1 Report

The article Manuscript ID: nanomaterials-1647514 with Title: A Sustainable Strategy for Solid-phase Extraction of Antiviral Drug from Environmental Waters by Immobilized Hydrogen Bond Acceptor. The authors are addressed the comments well and the updated version is looking good. I recommend accepting the manuscript.